# Effects of a Combined High-Intensity Interval Training and Resistance Training Program in Patients Awaiting Bariatric Surgery: A Pilot Study

**DOI:** 10.3390/sports7030072

**Published:** 2019-03-25

**Authors:** Inés Picó-Sirvent, Adolfo Aracil-Marco, Diego Pastor, Manuel Moya-Ramón

**Affiliations:** 1Department of Sport Sciences; Sport Research Centre, Miguel Hernández University, 03202 Elche, Spain; mpico@umh.es (I.P.-S.); fito@umh.es (A.A.-M.); dpastor@umh.es (D.P.); 2Instituto de Neurociencias, UMH-CSIC, Miguel Hernández University, 03550 San Juan de Alicante, Spain; 3Institute of Health and Biomedical Research (ISABIAL-FISABIO Foundation), Miguel Hernández University, 03010 Alicante, Spain

**Keywords:** morbid obesity, resistance training, HIIT, body composition, cardiometabolic risk

## Abstract

Bariatric surgery (BS) is considered the most effective treatment for morbid obesity. Preoperative body weight is directly associated with a higher surgical morbimortality and physical activity could be effective in the preparation of patients. The aim of this study is to determine the effects of a six-month exercise training program (ETP), combining high-intensity interval training (HIIT) and resistance training in patients awaiting BS. Six candidates awaiting BS (38.78 ± 1.18 kg·m^−2^; 38.17 ± 12.06 years) were distributed into two groups: the ETP group (experimental group (EG), n = 3) and a control group (CG, n = 3). Anthropometrical and blood pressure (BP), cardiorespiratory fitness and maximal strength were registered before and after the ETP. The EG participated in 93.25% of the sessions, showing reductions in body mass index (BMI) compared to the CG (34.61 ± 1.56 vs. 39.75 ± 0.65, *p* = 0.006, ANOVA). The inferential analysis showed larger effects on BMI, excess body weight percentage and fat mass, in addition to small to moderate effects in BP and the anthropometric measurements. Peak oxygen uptake normalized to fat-free mass showed likely positive effects with a probability of >95–99%. A six-month ETP seems to be a positive tool to improve body composition, cardiometabolic health, and fitness level in patients awaiting BS, but a larger sample size is needed to confirm these findings.

## 1. Introduction

Obesity is considered a worldwide epidemic, one of the greatest public health problems, and one of the primary death causes, both in the European Union and in the USA [1,2]; likewise, obesity is associated with cardiovascular disease risk factors, some types of cancer, type 2 diabetes mellitus, hypertension, hyperlipidemia, immune response affectation, greater morbidity and mortality state, and a considerable reduction of life expectancy [3,4]. In addition, the increase in the obesity rate and its associated diseases have been related to an increase in the variety of therapeutic interventions [5]. In cases of morbid obesity (body mass index (BMI) ≥ 40 kg/m^2^), bariatric surgery (BS) has been demonstrated to be the most effective treatment for the loss and maintenance of body weight, as well as for the improvement of comorbidities and mortality associated with this pathology [6]. Nevertheless, even though physical activity is recommended to optimize BS results, nowadays the majority of candidates for BS are sedentary and do not increase their physical activity levels after surgery [7].

The influence of physical activity in maintaining long-term weight loss has been known for years, but its role in the preparation of patients awaiting surgery is not well-known. It has been suggested that it could reduce anesthetic risk and improve postoperative recovery [8]. The higher the preoperative weight is, the higher the risk of morbidity and mortality is. Currently, a low-calorie diet is recommended in short-term preoperative weight loss interventions, as these diets seem to be related to a decrease in the liver size and intraabdominal fat mass which reduce the surgical risk [9]. However, the loss of fat-free mass (FFM) associated with these interventions is an unintended consequence of BS and it is expected to cause a decrease in resting energy expenditure (REE) that could be related to weight regain after surgery [10]. Consequently, it is necessary to implement programs that improve the preoperative conditions and reduce the risk factors in patients awaiting BS. Experimental studies show that exercise training program (ETP) physical activity programs after BS provide benefits for health and fitness level in individuals with morbid obesity [11,12,13]. Moreover, after a physical activity program lasting six months prior to BS, there were increases in the practice of postoperative physical activity [14]. Other studies in which preoperative ETPs were conducted combining aerobic training with resistance training or only applying aerobic or resistance training have been shown to be effective in improving body weight, physical condition, quality of life related to health, and they also help to improve their willingness to practice physical activity in candidates awaiting BS [3,8,15,16,17,18,19]. However, methodologies to prescribe a training workload are not well-defined in this population.

Several studies have consistently found that high-intensity interval training (HIIT) is useful to increase aerobic capacity and muscle mass, the benefits of which are similar to those obtained with moderate-intensity continuous training (MICT) [20,21,22,23,24]. Nevertheless, HIIT is a time-efficient methodology that allows improvements in cardiorespiratory fitness and work capacity, mitochondrial muscle biogenesis and GLUT-4 levels, insulin sensitivity, fasting glucose, HbA1c and reduces several cardiometabolic risk factors in overweight/obese populations [25,26,27].

In addition, to our knowledge, there are no studies that analyze the effects of applying an ETP to increase or maintain muscle mass combining HIIT and resistance training with progressive loads. Although it has been amply demonstrated that resistance training with progressive resistance is safe, effective, and potentially more valid than aerobic exercise in many groups of patients [19], there is little scientific evidence to determine the exercise type, training loads to apply and results derived from ETP in patients with type III obesity awaiting BS.

The main purpose of this study was to determine the effects of a structured ETP, combining HIIT and resistance training with progressive workloads on the anthropometric profile, cardiometabolic risk factors (CRF), cardiorespiratory fitness and strength levels of patients with morbid obesity that are candidates for BS.

## 2. Materials and Methods

### 2.1. Participants

Six patients awaiting BS voluntarily took part in this study after being recommended to participate by the obesity surgery medical team both from the University Hospital of Vinalopó and the University General Hospital of Elche. They were divided, depending on their suitability possibilities to participate in a training program, into an experimental group (EG) or a control group (CG) that followed the usual medical care indications, see Table 1. Patients were eligible if they were awaiting BS and led a sedentary lifestyle (less than one hour of structured exercise weekly). The exclusion criteria were to suffer from: (a) any cardiovascular disease; (b) asthma or chronic obstructive pulmonary disease; (c) hypothyroidism or (d) functional limitations to perform an ETP. All participants were carefully informed about the risks associated with the study, and they were asked to sign an informed consent based on the Helsinki Declaration and approved by the University Ethical Committee.

### 2.2. Procedure

All participants followed the usual presurgical care indications from their respective hospitals. The EG (n = 3) performed a six-month structured supervised ETP, in which the aerobic exercise they performed was carried out at high intensities and the resistance training was carried out with high workloads. The CG (n = 3) performed the same testing protocol, but without ETP.

The ETP was performed in the sports facilities of the University, under the direct supervision of sport sciences graduates. The weekly training frequency was established as two weekly sessions the first month up to four weekly sessions from the third month until the end of the program. The activities were individualized for each patient.

The EG and CG were tested a week before starting the ETP (E1), and six months after, at the end of the program (E3). In addition, the EG underwent an intermediate evaluation (E2) to check the evolution of the participants and to adapt the training loads. Body composition, cardiometabolic risk factors (CRF) and physical fitness were measured in a laboratory under controlled conditions. 

### 2.3. Anthropometry, Body Composition and CRF Measurements

Patients visited the laboratory between 0700 and 0900 hours, after 12 h of overnight fasting, and with an empty bladder [28]. Caffeine or alcohol consumption and exercise were forbidden in the 48 h prior to the test. Total body weight and body composition were measured by bioimpedance analysis (Tanita BC-420MA, Tanita, Tokyo, Japan), and body mass index was calculated and expressed as kg·m^−2^. Blood pressure was measured according to established recommendations [29] using a digital sphygmomanometer (Microlife WatchBP Home, Heerbrugg, Switzerland). Both weight and the waist and hip circumferences were measured using the ISAK (International Society for the Advancement of Kinanthropometry) protocol [30].

### 2.4. Cardiorespiratory Fitness Measurement

A two-phase protocol on a cycle ergometer (Technogym Bike Med, Technogym, Gambettola, Italy), adapted from a previous study [31] was performed to determine the peak oxygen uptake (VO_2peak_) and fat oxidation rates. Respiratory exchange was registered with an Oxycon Pro gas analysis system (Jaeger, Friedberg, Germany). Carbohydrate oxidation (CHO) and fat oxidation (FO) were calculated in the first phase, while the second phase was used to establish the VO_2peak_. The first phase consisted of performing a 4-min warm-up at 40 watts (W), followed by increases of 20 W every 3 min, maintaining a cadence of 60 rotations per minute (rpm). The first phase ended if the respiratory exchange ratio (RER) reached 1.0 or cadence values reduced to under 60 rpm. At this point, the second phase, consisting of 20 W increments each minute until volitional fatigue, started. To calculate the VO_2peak_, the average of the highest 30 s of peak oxygen was used and it was expressed in absolute values (VO_2peak abs_).

### 2.5. Muscle Strength Measurements

Maximal dynamic and isometric strength (MDS and MIS, respectively) of the quadriceps and hamstrings were assessed by an isokinetic dynamometer (Biodex System 4; Biodex Medical Systems, New York, NY, USA). Participants performed a 5-min warm-up on a cycle ergometer (Technogym Bike Med, Technogym, Gambettola, Italy) at an intensity of 60% maximal heart rate (HR_max_) before starting the test. After that, participants were seated on the dynamometer chair, with the chair back inclined at an angle of 85° between hip and back. Two Velcro straps were used to avoid pelvis and torso movements. The dynamometer rotation axis was aligned with the lateral femoral epicondyle, allowing movement in the sagittal plane. Three unilateral tests were performed with each limb, one isokinetic test and two isometric tests. The dominant limb was evaluated first, followed by the non-dominant limb, with a two-minute rest between tests and a three-minute rest between limbs.

*MDS.* Four sets of four concentric contractions (flexion-extension) of the knee were performed at an angular speed of 60°/s, with 90 s rest between repetitions. The knee motion range was 10°–105°. A first submaximal trial was performed to familiarize the participants with the protocol. The three following sets were performed at maximal individual effort, and participants were verbally encouraged to maintain their maximal effort in every contraction. The highest value of the last three sets was chosen to analyze the peak torque (N·m).

*MIS*. Participants completed a Maximum Voluntary Contraction (MVC) trial of four repetitions (the first set as a submaximal familiarization trial), with a fifteen-second rest between repetitions and were given verbal encouragement to perform at maximal effort for five seconds [32]. The angles used to assess the lower limbs were 105° for quadriceps and 75° for hamstrings.

### 2.6. Exercise Training Program (ETP)

A six-month ETP was performed by the EG, initially combining aerobic exercise at moderate intensities and resistance training to familiarise participants with physical exercise practice, see Table 2. The endurance training progressed from 60% to 95% VO_2peak_ intensities. The weekly frequency of the program was 2 days weekly (d/w) the first month, 3 d/w the second month, and 4 d/w from the third to the last month. The sessions lasted 60 minutes for the first two months and 70 minutes from the third to the last month. The endurance training days alternated MICT and HIIT. MICT sessions were performed from 60% to 85% HR_peak_, while HIIT sessions were performed based on the percentage of VO_2peak_. The HIIT programs were only performed on a cycle ergometer and consisted of intervals of 30 s of work interspersed by 30 s of active recovery at 30% VO_2peak_. The MICT was carried out on a treadmill or on a cycle ergometer.

Resistance training was carried out after HIIT in the same session, prescribed based on the percentage of one repetition maximum (1RM), which was determined using the Brzycki formula [33]. The exercises (latissimus dorsi, pectorals, quadriceps, hamstrings, gastrocnemius, deltoids, triceps, and biceps brachial) were performed using guided machines with weights that ranged between 50% of 1RM and 75% of 1RM. All the sessions finished with five stretching exercises in standing (gastrocnemius, quadriceps, hamstrings, pectorals and latissimus dorsi).

To provide an internal load measure and to control the training process, each participant’s rating of perceived exertion (RPE) was collected using the CR-10 Borg’s scale [34]. RPE was taken 30 min after the end of each session, in order to prevent difficult or easy exercises near the end of the session that could influence the overall rating of the session [35]. 

### 2.7. Statistical Analysis

The data are presented as mean (±SD). The normal distribution of all data series was verified by the Kolmogorov–Smirnov test. A one-factor ANOVA was used to check the changes produced in physical condition, body composition and cardiometabolic variables for each of the two testing moments (pre-program vs post-program). A t-test was performed for multiple within-group comparisons when ANOVA showed significant interaction effects among groups. 

All data were log-transformed for the analysis to reduce bias arising from non-uniformity error and then they were analyzed for practical significance using magnitude-based inferences [36]. The effect size (ES) was calculated using Cohen’s coefficient [37], which allowed the magnitude of the changes between the test and the post-test to be established based on the performance of the training session (i.e., ES of ≥ 0.2 = small; ≥ 0.5 = moderate; ≥ 0.8 = large magnitudes of change, respectively). Descriptors were used to interpret the probabilities of effects (clinical inferences based on threshold chances of harm and benefit of 0.5% and 25%) [38]. The probability that the true effects are harmful, trivial or beneficial was considered as the following: <1%, almost certainly not; 1–4%, very unlikely; 5–24%, unlikely or probably not; 25–74%, possibly or maybe; 75–94%, likely or probable; 95–99%, very likely; >99%, almost certainly [38]. Changes in mean were also included for each group (expressed in %) and their 90% confidence limits (CL).

Statistical analysis was performed with the SPSS package program (version 22, SPSS Inc., Chicago, IL, USA) and the null hypothesis was rejected at the 95% significance level (p ≤ 0.05). For clarity, all results are presented as positive improvements, so that negative and positive differences can be viewed in the same direction.

## 3. Results

### 3.1. Description of the Dynamic Loads Performed by the Experimental Group

The EG participated in 93.25 ± 6.5% of the sessions, without any injury or adverse event experienced. Internal load values (calculated as the product of total session time duration and intensity), the monotony and the strain index during each week of ETP are shown in Figure 1.

### 3.2. Anthropometry and Body Composition

After performing the ETP, ANOVA showed no statistically significant differences in all the variables evaluated in the EG, except in BMI and excess body weight percentage (EBW%), which had higher significant reductions than the CG (34.61 ± 1.56% vs 39.75 ± 0.65%, p = 0.006, and 27.00 ± 4.06% vs 37.10 ± 1.02%, p = 0.014, respectively), as shown in Figure 2. 

T-test results are shown in Table 3 for the CG and EG, while among-group differences in the variables considered as the most relevant ones are represented in Figure 3. The EG shows a larger ES on BMI, EBW (kg, %) and fat mass (kg, %), which is not observed in the CG. In addition, all the anthropometric measurements showed small to moderate ES except FFM (kg) values. The inferential analysis showed unclear effects in BMI, EBW (kg, %), fat mass percentage (FM%) and visceral fat with a probability of ≥80–95%. However, likely positive effects with a probability of >95–99% were shown in the fat-free mass percentage (FFM%). Finally, two correlations between fat mass (kg) and FFM%, FM% and hip circumference were observed (r = −0.875 and r = 0.822, respectively). Qualitative inference and likelihood (%) of the effects being positive/trivial/negative for each of the groups could be consulted in Appendix A.

### 3.3. Physical Fitness and Cardiometabolic Risk Factors

After performing the ETP, significant differences among groups were found in MDS normalized to FFM (MDSFFM) in the non-dominant (ND) limb hamstrings (EG = 138.34±3.29 N·m·FFM^−1^ vs CG = 106.68 ± 6.77 N·m·FFM^−1^; p = 0.002), as well as in MIS in the ND hamstrings normalized to FFM (EG = 145.68 ± 8.41 N·m·FFM^−1^ vs CG = 106.98 ± 10.85 N·m·FFM^−1^; p = 0.008) and in the ND quadriceps (EG = 151.73± 16.01 N·m vs CG = 121.02 ± 8.40 N·m; p = 0.042).

T-test results are shown in Table 4 for the CG and EG, respectively. Both the EG and the CG showed improvements in diastolic blood pressure (DBP), VO_2peak_ when it was normalized to FFM (VO_2peak abs_/FFM) and ND hamstring MDS level, but only the EG improved hip circumference, systolic blood pressure (SBP) and MDS level in the ND quadriceps. However, the EG showed small and moderate effects on MDS in the ND lower limb both in quadriceps (dQ = 0.35, likely beneficial) and hamstrings (dH = 0.56, likely beneficial), while inferior results were found in the CG for the same variables (dQ = 0.02, possibly trivial; dH = 0.32, likely beneficial). Qualitative inference and likelihood (%) of the effects being positive/trivial/negative for each of the groups can be consulted in Appendix A.

## 4. Discussion

Our pilot study suggests that a six-month ETP, combining long-term HIIT with resistance training from 55% to 75% 1RM seems to be a positive intervention to improve body composition, CRF and physical condition in patients awaiting BS. Despite the limitations associated with our small sample size, this pilot study could be useful as a starting point to design future research with similar physical activity interventions. The ETP could be feasible since subjects participated in >90% of the total supervised sessions proposed, which resulted in practical significant improvements in most of the anthropometrical variables, SBP, DBP, cardiorespiratory fitness and maximal strength level in the non-dominant lower limb. To our knowledge, this study is the first to examine the effects of an ETP combining endurance and resistance training with a programed and monitored workload in this population.

The results related to changes in FM observed in the EG could explain significant reductions in BMI and EBW% (p < 0.05). Participants in the EG showed FM and visceral fat reductions after the ETP, while FFM remained constant. Therefore, FFM% increased after the ETP. Conservative weight loss programs aim to lose from 5% to 10% body weight to be able to consider a reduction as clinically significant to decrease surgical risk [39]. In this pilot study, the EG showed a decrease close to 6% in EBW% (33.55 ± 3.89 vs 27.00 ± 4.06, p = 0.052). Consequently, our ETP seems to be a positive tool that could be analyzed and added to traditional weight loss programs to complement the effects associated with pharmacological, psychological and nutritional treatments in patients awaiting BS. In fact, a recent study showed that a twice-weekly low-intensity physical activity program encouraged individuals to adopt a more active lifestyle, both with and without the aid of support group therapy for lifestyle modification [40]. Therefore, multidisciplinary treatments that include physical activity could be the key to optimize not only weight loss parameters before surgery, but also the surgical risk and health status of patients, helping them to adopt more physically active lifestyles.

There are no standardized guidelines to establish recommendations of physical activity in obese patients awaiting BS. Previous studies [4,17,41] have performed aerobic exercise combined with resistance training which reported changes in body composition and anthropometrical variables. Our pilot study is characterized by programming and monitoring workloads over a six-month period, showing higher reductions in waist and hip circumferences, BMI, body weight, FM and visceral fat percentages than results reported by similar studies [8,17,42]. These results could be explained by two reasons: a) The longer duration of our program (84 training sessions in total), and b) the individualization of training prescription according to percentages of their VO_2peak_ and 1RM tests. In addition, none of these interventions compare their results with a CG that followed the usual care indications, so it is difficult to interpret the effect that is directly associated with physical activity. Only Baillot et al. [42] compared the effects of a 12-week intervention with a CG, the results of which did not present changes in body composition and CRF. Conversely, our data show differences with practical significance in body composition, among patients that performed a structured ETP or followed the usual medical care; therefore it can be guaranteed that changes are not a consequence of a traditional BS treatment preparation exclusively.

Small to moderate changes were observed in most of the CRF and physical condition variables after performing an ETP. Both the EG and the CG showed improvements in some maximal muscle strength variables. However, although the EG showed higher changes in VO_2peak abs_/FFM, the CG improved VO_2peak abs_ as well, so no influence of the previous experience of participants during E1 could be implied. Because severely obese adults often suffer from chronic knee pain, breathing arrhythmia and exercise intolerance [43], patients could be limited to performing a maximal test to prevent injuries or negative experiences. Thus, these results could be explained by the absence of learning sessions of both isokinetic and cardiorespiratory test protocols. Therefore, these sessions could be an alternative so that patients get to know each protocol, and consequently, to check that the results obtained are consistent and reliable.

Our results in MDS levels can be explained by excessive intra- and intermuscular fat accumulation which characterizes our population and influences both the force generation and the functional capacity of a skeletal muscle negatively. Although our ETP is related to FM% and fat visceral reductions, the electrical bioimpedance method cannot discriminate the body area in which greater FM has been reduced. A recent study [44] has suggested that FFM, mainly in the lower limbs, is crucial to increase the absolute MVC torques and, consequently, to prevent losses of muscle strength and functional limitations. Therefore, it could be because our participants did not show significant differences in MDS and MIS after performing an ETP as FFM remained constant and changes in FFM% were a direct consequence of body weight reduction due to FM decrease.

All participants were sedentary adults with no previous experience in practicing physical activity. This is particularly interesting since one of the most relevant practical problems for exercise professionals is the amount of physical activity required to reach specific outcomes without a negative impact. Although our ETP was performed with high workloads, no significant changes in training load were registered from the third to the sixth month of the ETP, see Figure 1. This could imply that patients with severe obesity likely need approximately eight weeks, see Figure 1, to adapt to the systematic practice of physical activity. In addition, the strain of training was well-controlled through interspersing combined high-intensity training sessions (HIIT and resistance training) with MICT sessions from the second to the sixth month, which could explain why no participants suffered any injury during the ETP.

## 5. Conclusions

In summary, our data suggest that a six-month ETP could be a positive and safe intervention to support traditional weight loss treatments for patients awaiting BS. However, this pilot study is not enough to establish specific guidelines. Future studies with a larger sample size are needed to confirm these benefits which are apparently associated with an ETP before BS. In addition, post-surgery data would be interesting to be able to assess the real effectivity to reduce surgical risk and improve the recovery process. 

## Figures and Tables

**Figure 1 sports-07-00072-f001:**
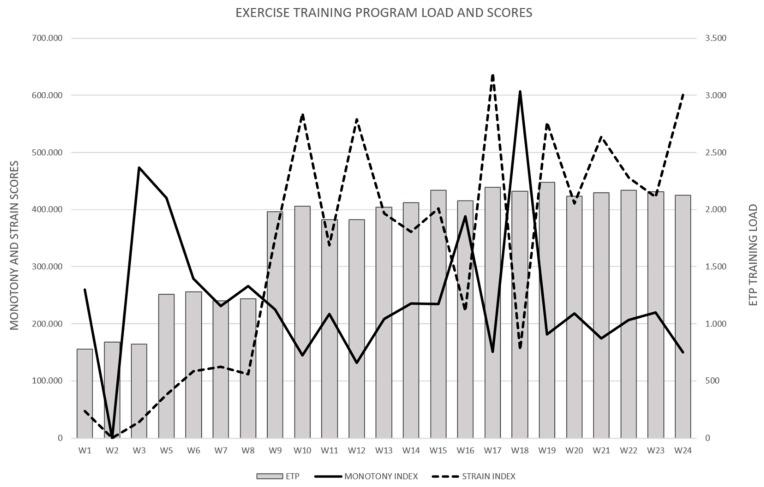
Load dynamics and scores during each week of ETP. W = week; ETP = exercise training program. * Week 4 does not show load values because 1RM tests were performed.

**Figure 2 sports-07-00072-f002:**
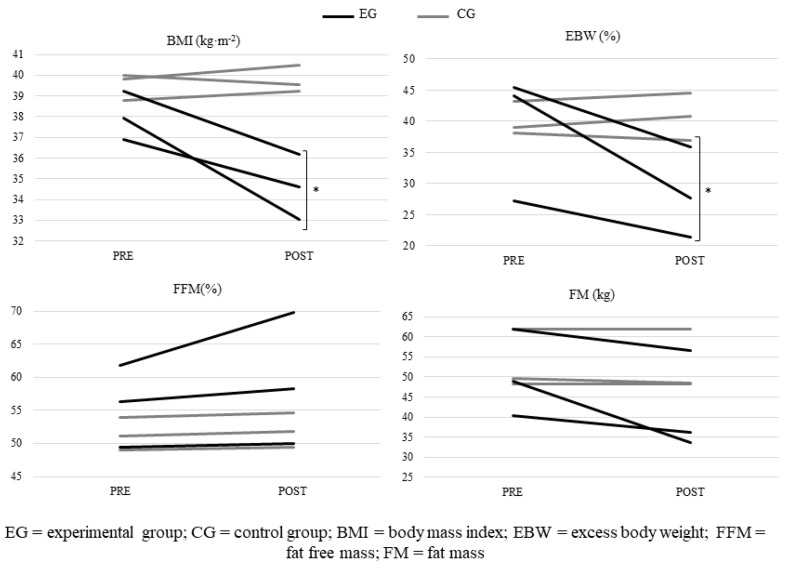
Changes in each participant in body composition variables after vs before ETP (* *p* < 0.05). There was no significant difference between the two groups for values observed before training.

**Figure 3 sports-07-00072-f003:**
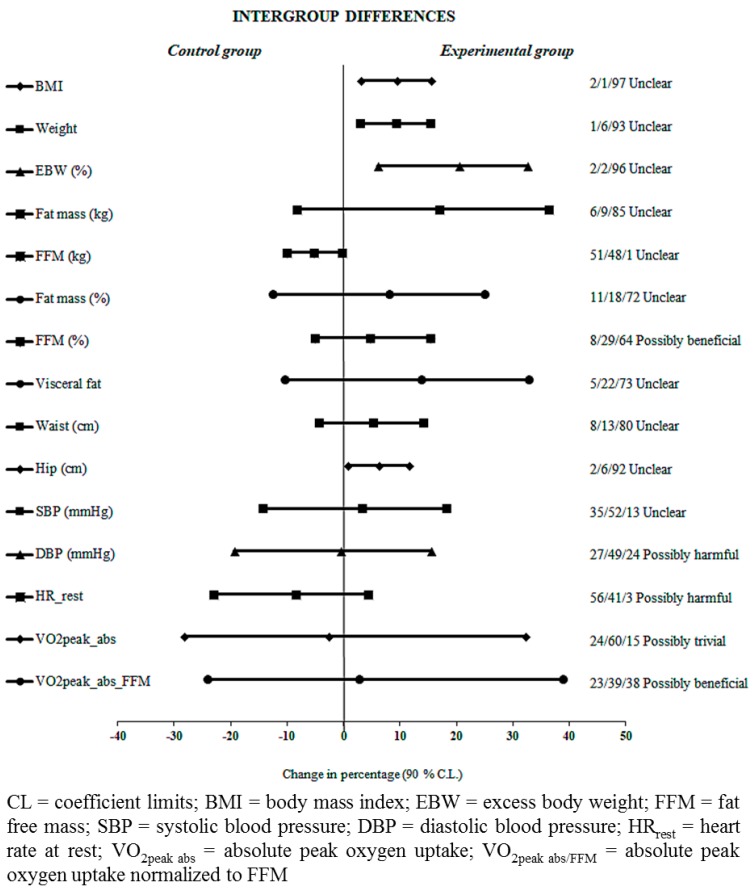
Changes in anthropometric measurements, cardiometabolic risk factors and physical fitness variables among groups.

**Table 1 sports-07-00072-t001:** Baseline characteristics of participants before starting the exercise training program (ETP).

Variables	Total (n = 6)	EG (n = 3)	CG (n = 3)
Female (n)	5	2	3
Age (years)	38.17 ± 12.06	39.67 ± 10.21	36.67 ± 15.88
BMI (Kg·m^−2^)	38.78 ± 1.18	38.02 ± 1.16	39.54 ± 0.64
Weight (Kg)	111.83 ± 14.10	114.43 ± 19.11	109.23 ± 10.57
Fat mass (%)	46.40 ± 4.89	44.17 ± 6.21	48.63 ± 2.51
FFM (%)	53.60 ± 4.89	55.83 ± 6.21	51.37 ± 2.51
Visceral fat (%)	14.00 ± 3.88	15 ± 5.51	13.00 ± 2.08
VO_2peak abs_ (L·min^−1^)	2.20 ± 0.86	2.62 ± 1.11	1.78 ± 0.29
VO_2peak abs/FFM_ (mL·FFM^−1^·min^−1^)	35.77 ± 7.14	39.92 ± 8.01	31.61 ± 3.40
Systolic blood pressure (mmHg)	131.89 ± 27.39	144.22 ± 37.39	119.56 ± 4.53
Diastolic blood pressure (mmHg)	77.94 ± 13.48	83.22 ± 18.68	72.67 ± 4.63

Values are mean ± SD. ETP = exercise training program; EG = experimental group; CG = control group; Kg = kilograms; m = metres; BMI = body mass index; FFM = fat free mass; VO_2peak abs_ = absolute peak oxygen uptake.

**Table 2 sports-07-00072-t002:** ETP schedule (EG).

	Month	1	2	3	4	5	6
**MICT**	Weekly frequency (sessions/week)	2	1	2	2	2	2
	Volume (min)	35	50	50	50	50	50
	Intensity (% HR_peak_)	60–70	65–75	70–80	70–85	70–85	70–85
**HIIT**	Weekly frequency (sessions/week)		2	2	2	2	2
	Volume (min)		20	20	20	20	20
	Intensity (% VO_2peak_)		60–70	70–80	75–85	80–90	80–95
**Resistance training**	Weekly frequency (sessions/week)	2	2	2	2	2	2
	Volume (series × exer. × rep.)	1 × 5 × 20	1 × 7 × 20	4 × 4 × 15	4 × 4 × 12	4 × 4 × 10	4 × 4 × 10
	Intensity (% 1RM)	55	60	65	70	75	75
**Stretching training**	Weekly frequency (sessions/week)	2	3	4	4	4	4
	Volume (series × exer.)	1 × 5	1 × 5	1 × 5	1 × 5	1 × 5	1 × 5
	Duration (min)	1	1	1	1	1	1

ETP = exercise training program; EG = experimental group; MICT= moderate-intensity continuous training; HIIT = high-intensity interval training; % HR_peak_ = peak heart rate percentage; min = minutes; % VO_2peak_ = peak oxygen uptake percentage; exer. x rep. = number of exercises performed and number of repetitions by exercise; % 1RM = percentage over a maximum repetition.

**Table 3 sports-07-00072-t003:** Descriptive characteristics (mean ± SD) on anthropometric measurements before and after the ETP for each of the groups.

	EG (n = 3)			CG (n = 3)		
Variables	Pre	Post	ES (d) (90% CL)	Pre	Post	ES (d) (90% CL)
**BMI (kg·m^−2^)**	**38.02 ± 1.16**	**34.61 ± 1.56 ***	3.30 (1.05; 5.54)	39.54 ± 0.64	39.75 ± 0.65	**−0.19 (−1.03; 0.66)**
Weight (kg)	114.43 ± 19.11	103.87 ± 14.80	0.57 (0.18; 0.57)	109.23 ± 10.57	109.87 ± 11.43	−0.03 (−0.18; 0.11)
EBW (kg)	38.83 ± 10.14	28.27 ± 7.27	2.71 (0.79; 4.62)	40.09 ± 2.70	40.72 ± 3.74	−0.12 (−0.68; 0.43)
**EBW (%)**	**33.55 ± 3.89**	**27.00 ± 4.06 ^†^**	4.42 (1.19; 7.65)	36.77 ± 1.04	37.10 ± 1.02	**−0.18 (−1.00; 0.63)**
Fat mass (kg)	50.48 ± 10.87	42.16 ± 12.52	0.81 (−0.30; 1.91)	53.25 ± 7.49	52.91 ± 7.91	0.03 (−0.08; 0.14)
**Fat mass (%)**	44.17 ± 6.21	40.63 ± 9.94	**−1.08 (−1.15; 3.31)**	**48.63 ± 2.51**	**48.03 ± 2.65 ***	0.14 (0.06; 0.21)
FFM (kg)	63.95 ± 13.83	61.71 ± 14.39	−0.32 (−0.70; 0.07)	55.98 ± 3.80	56.96 ± 4.40	0.14 (−0.05; 0.33)
**FFM (%)**	55.83 ± 6.21	59.37 ± 9.94	**0.66 (−0.48; 1.80)**	**51.37 ± 2.51**	**51.97 ± 2.65 ***	0.14 (0.08; 0.19)
Waist (cm)	120.27 ± 4.07	114.26 ± 5.68	0.37 (−0.31; 1.05)	115.62 ± 9.10	116.02 ± 8.18	−0.03 (−0.21; 0.16)
**Hip (cm)**	**130.57 ± 10.76**	**122.27 ± 13.63 ^†^**	0.69 (0.12; 1.26)	134.04 ± 7.64	133.70 ± 8.72	0.03 (−0.14; 0.20)
WHR (cm)	0.93 ± 0.09	0.94 ± 0.06	−0.10 (−0.54; 0.35)	0.86 ± 0.08	0.87 ± 0.08	−0.04 (−0.10; 0.02)
**Visceral fat (%)**	14.67 ± 5.51	12 ± 2.65	**0.62 (−0.24; 1.47)**	12.67 ± 2.08	12.33 ± 1.53	0.08 (−0.16; 0.32)

PAP = physical activity program; EG = experimental group; CG = control group; CL = coefficient limits; BMI = body mass index; EBW = excess body weight; FFM = fat free mass; WHR = waist-to-hip ratio. * Statistical significant intragroup differences (*p* < 0.05, paired t-test). ^†^ (*p* = 0.052; *p* = 0.059, paired t-test).

**Table 4 sports-07-00072-t004:** Descriptive characteristics (mean ± SD) on cardiometabolic risk factors and physical fitness variables before and after the PAP, for each of the groups.

	EG (n = 3)			CG (n = 3)		
Variables	Pre	Post	ES (d) (90% CL)	Pre	Post	ES (d) (90% CL)
SBP (mmHg)	144.22 ± 37.39	139.28 ± 28.76	0.41 (−2.02; 2.84)	119.56 ± 4.53	120.61 ± 8.92	−0.11 (−2.09; 1.86)
DBP (mmHg)	83.22 ± 18.68	79.83 ± 18.37	0.37 (−0.53; 1.28)	72.67 ± 4.63	69.39 ± 6.09	0.42 (−1.23; 2.06)
HR_rest_ (bpm)	56.33 ± 12.34	62.00 ± 11.53	−0.26 (−0.54; 0.03)	65.67 ± 13.58	66.67 ± 13.32	−0.04 (−0.32; 0.23)
VO_2peak abs_ (L·min^−1^)	2.62 ± 1.11	2.69 ± 1.04	0.11 (−0.12; 0.35)	1.78 ± 0.29	1.89 ± 0.37	**0.21 (** **−** **0.85; 1.28)**
VO_2peak abs/FFM_ (mL*·FFM^−1^·min^−1^)	39.92 ± 8.01	42.58 ± 6.24	**0.38 (** **−** **0.23; 0.99)**	31.61 ± 3.40	33.07 ± 4.45	**0.23 (** **−** **1.29; 1.76)**
MDS_D-Q_ (N·m)	151.84 ± 36.10	155.16 ± 43.38	0.05 (−0.36; 0.45)	144.12 ± 21.46	149.50 ± 18.35	**0.14 (** **−** **0.16; 0.45)**
MDS_D-H_ (N·m)	86.53 ± 24.87	84.54 ± 17.37	−0.02 (−0.53; 0.48)	66.06 ± 18.67	59.93 ± 14.97	−0.17 (−0.40; 0.07)
MDS_ND-Q_ (N·m)	149.88 ± 47.94	161.74 ± 54.86	**0.35 (0.13; 0.58)**	141.14 ± 15.73	142.03 ± 19.00	0.02 (−0.44; 0.49)
MDS_ND-H_ (N·m)	75.81 ± 14.93	85.66 ± 21.93	**0.56 (** **−** **0.21; 1.32)**	57.03 ± 6.42	60.81 ± 6.69	**0.32 (0.06; 0.57)**
MIS_D-Q_ (N·m)	141.90 ± 19.18	139.77 ± 21.19	−0.07 (−0.30; 0.16)	138.76 ± 19.35	130.11 ± 3.89	−0.24 (−1.02; 0.54)
MIS_D-H_ (N·m)	95.83 ± 23.39	90.53 ± 12.71	−0.12 (−0.69; 0.45)	72.16 ± 15.02	62.48 ± 14.31	−0.37 (−1.20; 0.45)
**MIS_ND-Q_ (N·m)**	147.50 ± 3.47	151.73 ± 16.01	0.11 (−0.48; 0.69)	133.61 ± 17.80	121.02 ± 8.40	−0.40 (−0.90; 0.10)
MIS_ND-H_ (N·m)	90.30 ± 15.95	89.10 ± 15.30	−0.02 (−0.32; 0.28)	61.71 ± 19.85	61.18 ± 10.31	−0.03 (−0.52; 0.46)

PAP = physical activity program; EG = experimental group; CG = control group; CL = coefficient limits; SBP = systolic blood pressure; DBP = diastolic blood pressure; HR_rest_ = heart rate at rest; VO_2peak abs_ = absolute peak oxygen uptake; VO_2peak abs/FFM_ = absolute peak oxygen uptake normalized to FFM; MDS = maximal dynamic strength; D = dominant; ND = non-dominant; Q = quadriceps; H = hamstrings; MIS = maximal isokinetic strength. * Statistical significant intragroup differences (*p* < 0.05).

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
