# Peer review of "Effects of a Combined High-Intensity Interval Training and Resistance Training Program in Patients Awaiting Bariatric Surgery: A Pilot Study"

_sports, 2019, doi:10.3390/sports7030072_

Round 1

Reviewer 1 Report

General comments -

Quite a nice study, small but important. Some minor issues re. written English, please consider a native English speaking proof reader. Did you monitor any medication use throughout the duration of the study? I think there needs to be more emphasis on the feasibility of 6 months of personalised PAP given that this is a very small pilot study.

Please include combined HIIT & (MICT?) RT in title

Abstract - would like to see the actual reduction in BMI

Intro - nice story leading to intervention

Consider including the following references:

Jelleyman et al (2015) Obesity Reviews 16(11)

Herring et al (2017) Obesity Reviews 41(6)

Methods - 

Please add how and where from the participants were recruited, as well as how many were excluded based on exclusion criteria as this will help to justify your small sample size.

Is Type 2 diabetes included in exclusion criteria? Did any of your patients have T2DM?

More info is required on the PAP's. You say in the abstract that this is a combined HIIT & RT PAPs but in the methods you say you used MICT as well. HIIT protocols should also include info about interval and rest length and intensity. This may need to be included in Supplementary materials.

Refer to Table 2 earlier in the PAP section

Figure 1 should be in the results

Please provide a reference for the strength of effect sizes and "true effects"

Results - 

Table 1 - Add N to Sex row.

Please list "all the variables evaluated"

Please add units to actual results (i.e. 34.61+/- 1.56kg.m^2)

Line 202 - should "among-group" be "within group"?

Figure 2 should appear after it is referred to

Figures 3 & 4 are very busy. Recommend removing some variables and perhaps separating body composition and CRF outcomes. Also would be good to present EG & CG in the same table. Could remove chances data and put in supplementary material.

Line 208-210 - what were the correlations and why are they relevant. Looks like phishing to me

I think that Tables 3&4 and figures 2 & 3 essentially show the same thing. Consider refining.

Line 216 - is this the first time you have referred to MDS/MIS? If not, remove non-abbreviated version and ensure this is defined on first use.

Please provide some actual results in the "Physical Fitness..." section

Adherence, attendance and injury/adverse event info would be useful

Discussion - 

Line 232 - future research

L242 - Not sure this is a logical conclusion from the previous sentence. Consider revising.

L275 - E1?

Author Response

Answers to reviewer 1

Dear Reviewer:

We appreciate the time you devoted to reading our manuscript and helping us to craft an improved version.  We are pleased to clarify your concerns, which we believe will improve the impact and quality of our work.  Please find below our responses to your observations. We have made a concerted attempt to systematically address the specific concerns raised for this revision and we have highlighted the alterations in the manuscript for convenience.  Additionally, we have carefully read the manuscript and improved the writing style and clarity, and altered any grammatical mistakes.

1)     Some minor issues re. written English, please consider a native English speaking proof reader.

We attach a proof reading certificate (pdf file).

2)     Did you monitor any medication use throughout the duration of the study?

Comment acknowledges. There was one participant with hypertension in each group who followed treatment.

3)     More emphasis on the feasibility of 6 months of personalised PAP given that this is a very small pilot study.

Comment acknowledges. We have tried to show the feasibility of personalised PAP throughout participants’ attendance.

4)     Please include combined HIIT & (MICT?) RT in title.

Comment acknowledges. We propose modifying as “Effects of a combined high intensity interval training and resistance training programme in patients awaiting bariatric surgery: a pilot study”.

5)     Abstract - would like to see the actual reduction in BMI.

Line 14: added mean and SD values both EG and CG, separately.

6)     Consider including the following references in the introduction: Jelleyman et al (2015) Obesity Reviews 16(11), Herring et al (2017) Obesity Reviews 41(6).

Line 63: reference included. We are grateful for your suggestion.

Comment acknowledges. We have not found the following reference: Herring et al (2017) Obesity Reviews 41(6). Instead of it, we know the following one: Herring et al (2017) International Journal of Obesity, 41(6). However, although it has considered as very interesting results, our pilot study is related to patients awaiting bariatric surgery and this RCT analyse the effects of an exercise programme in patients undergone bariatric surgery.

7)     Please add how and where from the participants were recruited, as well as how many were excluded based on exclusion criteria as this will help to justify your small sample size.

Comment acknowledges. Hospitals which collaborated with our project referred the patients to the training programme. Due to the fact that the clinical teams knew our exclusion criteria, only patients that could be enrolled were referred to us. The small sample size can be justified because: 1) some patients who were informed about our research project by medical team never contacted us because of personal issues (i.e., not living in Elche, lack of interest), and 2) each hospital had their own surgical calendar and, in case of more severe, urgent surgeries -such as cancer surgeries-, medical team explained us that bariatric surgeries would be interrupted.

8)     Is Type 2 diabetes included in exclusion criteria? Did any of your patients have T2DM?

Comment acknowledges. Type 2 diabetes was not included in our exclusion criteria, but none of our participants had it. On the contrary, there were one participant in each group with hypertension.

9)     More info is required on the PAP's. You say in the abstract that this is a combined HIIT & RT PAPs but in the methods you say you used MICT as well. HIIT protocols should also include info about interval and rest length and intensity.

Comment acknowledges. The aim of alternate MICT and combined training sessions (HIIT and RT) is to facilitate recovery between high-intensity sessions. Moreover, the first month sessions combined MICT and RT to familiarise participants with physical exercise practice.

Line 153: we have added the last explanation regarding the first month.

Line 159: we have included HIIT protocol carried out.

Line 163: we have added more information briefly to clarify combined sessions.

10) Refer to Table 2 earlier in the PAP section.

Comment acknowledges. We’re not sure to understand properly your suggestion. Table 2 is referred at the end of the first sentence of the paragraph, when PAP is starting to be treated.

11) Figure 1 should be in the results.

Comment acknowledges. We thank your suggestion. We have modified the structure according to it, adding a new subheading in the results.

12) Please provide a reference for the strength of effect sizes and "true effects".

Line 196: reference has been provided.

13) Table 1 - Add N to Sex row.

Comment acknowledges. We have expressed the sample as a number rather than as a percentage.

14) Please list "all the variables evaluated".

Comment acknowledges. We don’t understand clearly your comment. All the evaluated variables can be found at Tables 3 and 4.

15) Please add units to actual results (i.e. 34.61+/- 1.56kg.m^2).

Tables 3 and 4, and lines 227-230: units added.

16) Line 202 - should "among-group" be "within group"?

Comment acknowledges. It means differences among EG and CG as a result of 1-factor ANOVA.

17) Figure 2 should appear after it is referred to

Comment acknowledges. That suggestion has been applied to improve the manuscript.

18) Figures 3 & 4 are very busy. Recommend removing some variables and perhaps separating body composition and CRF outcomes. Also would be good to present EG & CG in the same table. Could remove chances data and put in supplementary material.

According to the comments of other referees, as well as yours, we have modified tables 3 and 4 to improve their understanding, and part of the data have been added to an extra table sent as a supplementary material. The new tables 3 and 4 show within-group mean values and ES of anthropometry and physical condition variables separately. We are grateful for your suggestion.

19) Line 208-210 - what were the correlations and why are they relevant. Looks like phishing to me.

Comment acknowledges. We consider these correlations relevant because patients awaiting bariatric surgery are usually asked for changing their habits to lose body weight, but usually it doesn’t be fulfilled healthy. According to Vázquez-Velázquez (2018), achieving a successful weight loss 1 year after bariatric surgery implies, in part, to lose a higher FM% but maintaining FFM. If patients achieve these results before the surgical interventions, we strongly consider that it will likely maintain healthy habits after being undergone surgery as well.

Likewise, these correlations show healthy effects on body composition associated with including a high demanding physical activity programme, combining concurrent training with MICT, in daily life of severe obese patients.

20) I think that Tables 3&4 and figures 2 & 3 essentially show the same thing. Consider refining.

Comment acknowledges. We consider that each figure and table show different information.

Because of our small sample size, we think that Figure 2 is relevant to take into consideration how the same physical exercise intervention affects differently each participant at the individual level, since changes that we can see throughout the mean value (Table 3) are the result of the effects of the programme in every participant in EG, not observed in the CG.

At Figure 3, we show intergroup comparison.

21) Line 216 - is this the first time you have referred to MDS/MIS? If not, remove non-abbreviated version and ensure this is defined on first use.

Comment acknowledges. Your remark is correct, we define these variables in line 129, so we have only used abbreviations now. We are grateful for your review.

22) Please provide some actual results in the "Physical Fitness..." section.

Lines 227-230: MDS in non-dominant limb for quadriceps and hamstrings has been added. After revising our database, we have also corrected a writing mistake detected.

23) Adherence, attendance and injury/adverse event info would be useful.

Line 224: information about attendance and injury/adverse event has been added.

24) Line 232 - future research.

Line 244: we have changed the word.

25) L242 - Not sure this is a logical conclusion from the previous sentence. Consider revising.

Line 265-268: we have tried to clarify this argument.

Comment acknowledges. Our main message is that, as a consequence of our programme, a FM loss is achieved while FFM is maintained.

26) L275 - E1?

Comment acknowledges. This abbreviation is defined in line 100 as baseline test, a week before starting the PAP.

Reviewer 2 Report

A concept confusion seems to be present in the manuscript, namely, HIIT and physical activity and exercise training. I suggest to change the “PAPs” physical activity program to other words as you studied the effects of exercise training program and not physical activity. From the conceptual point of view, the terms are not well applied. Please consider the suggestion or simply.

Line 55-57 “High intensity interval training (HIIT) is useful to increase aerobic capacity and muscle mass, the benefits of which are similar to those obtained with moderate intensity continuous training (MICT)”. I´m not in agreement with this sentence since HIIT is a high-intensity interval training where intensity is the main factor of the benefits found associated with this type of exercise training. I suggest to go deep on demonstrating why HIIT has potentially positive effects on health-related features.

Why asthma is an exclusion criteria?

The control group was composed mainly by female individuals?  Is there any reason for that? “They were divided, depending on their possibilities to participate in a training programme”? Is it not a randomized?

How do you monitored control group? They went to the University facilities or just stay at home? Otherwise, you have placebo effect.

Line 97- “and at the end of the program” should be without parentheses.

Table 2 contain a lot of information. I do not understand the probability data.

I suggest to put out the data simply and understandable at first sight.  

Author Response

Dear Reviewer:

We appreciate the time you devoted to reading our manuscript and helping us to craft an improved version.  We are pleased to clarify your concerns, which we believe will improve the impact and quality of our work.  Please find below our responses to your observations. We have made a concerted attempt to systematically address the specific concerns raised for this revision and we have highlighted the alterations in the manuscript for convenience.  Additionally, we have carefully read the manuscript and improved the writing style and clarity, and altered any grammatical mistakes.

Answers to reviewer 2

1)     A concept confusion seems to be present in the manuscript, namely, HIIT and physical activity and exercise training. I suggest to change the “PAPs” physical activity program to other words as you studied the effects of exercise training program and not physical activity. From the conceptual point of view, the terms are not well applied. Please consider the suggestion or simply.

Comment acknowledges. According to your suggestion, we have modified the title as: “Effects of a combined high intensity interval training and resistance training programme in patients awaiting bariatric surgery: a pilot study”, and used the proper terms throughout the manuscript.

2)     Line 55-57 “High intensity interval training (HIIT) is useful to increase aerobic capacity and muscle mass, the benefits of which are similar to those obtained with moderate intensity continuous training (MICT)”. I´m not in agreement with this sentence since HIIT is a high-intensity interval training where intensity is the main factor of the benefits found associated with this type of exercise training. I suggest to go deep on demonstrating why HIIT has potentially positive effects on health-related features.

Line 58: we have added some information regarding the effects of HIIT and MICT obtained in several interventional studies.

Line 63: we have included another reference related to the improvements of HIIT on cardiometabolic risk factors associated with type-2 diabetes.

3)     Why asthma is an exclusion criteria?

Comment acknowledges. To ensure the validity of maximal cardiorespiratory fitness test and, consequently, a reliable endurance training prescription both in HIIT and in MICT, we decided to include exclusively patients without any cardiorespiratory disease. All of our cardiorespiratory fitness test were carried out using an Oxycon Pro gas analysis system.

4)     The control group was composed mainly by female individuals?  Is there any reason for that? “They were divided, depending on their possibilities to participate in a training programme”? Is it not a randomized?

Comment acknowledges. Yes, our CG was composed only by women. Actually, 83% of the participants were female. Morbid obesity is much more prevalent in women than men (The Non-Communicable Disease Risk Factor Collaboration, 2017; Hruby & Hu, 2015). In addition, due to the fact that participating in the programme was voluntary and implied going to the sports facilities of University, participants who were uncapable of attending to regular sessions (mainly, lack of transport or time incompatibility with sessions), were assigned to the CG. Given that the majority of the participants were female, our CG randomly was composed only by women.

5)     How do you monitored control group? They went to the University facilities or just stay at home? Otherwise, you have placebo effect.

Comment acknowledges. All testing protocols were carried out in our training laboratory, in the same environment and conditions of EG, supervised by sport science professionals. However, they were not monitored with any intervention added to the regular advice of being physically active while waiting for bariatric surgery.

We considered that, due to the fact that participants in the CG were not constantly monitored, placebo effect could not be present.

6)     Line 97- “and at the end of the program” should be without parentheses.

Line 101: it has been modified.

7)     Table 2 contain a lot of information. I do not understand the probability data. I suggest to put out the data simply and understandable at first sight.

According to the comments of other referees, as well as yours, we have modified tables 3 and 4 to improve their understanding, and part of the data have been added to an extra table sent as a supplementary material. The new tables 3 and 4 show within-group mean values and ES of anthropometry and physical condition variables separately. We are grateful for your suggestion.

In addition, to interpret probability data ES and qualitative inference should be consulted simultaneously, analysing if coefficient limits have the 0 included.

Reviewer 3 Report

The present pilot study is well written and shows interesting findings in obese subject that awaiting bariatric surgery.  Although the results are promising, I do have some comments.

The present study studies an exercise program for obese patients. A major question here is how novel this is, as there are many studies that show beneficial effect for overweight or obese subjects, even with a much larger sample size. And awaiting surgery is not really different from subjects that are obese and not awaiting for surgery.

The abstract is straight forward, but lacks some patient characteristics. More information of the patients is warranted

The present training period is 6 months. How long does it generally take before subject ongoing surgery? Would 6 months be representative? Would the effects also be present in much shorter periods?

The authors rightfully describe that caloric restriction may be useful to lose weight. Is there any nutritional data available? And have the authors considered implementing a nutritional intervention for this group?

The introduction is very long, I would advise to shorten the introduction to improve readability.               

The results are well presented and described. However, there is an over presentation of the data. For example, lean body results are presented 3 times. In the table, individuals participant in a figure and in figure 3. The latter figure 3 I really like.

Is there any habitual physical activity data available. The latter is important to describe the energy expenditure which may explain the weight loss. What is the energy deficit that allows the ~9kg difference?

Author Response

Dear Reviewer:

We appreciate the time you devoted to reading our manuscript and helping us to craft an improved version.  We are pleased to clarify your concerns, which we believe will improve the impact and quality of our work.  Please find below our responses to your observations. We have made a concerted attempt to systematically address the specific concerns raised for this revision and we have highlighted the alterations in the manuscript for convenience.  Additionally, we have carefully read the manuscript and improved the writing style and clarity, and altered any grammatical mistakes.

Answers to reviewer 3

1)     The abstract is straight forward, but lacks some patient characteristics. More information of the patients is warranted.

Comment acknowledges. Regarding the extension of the abstract stablished by the journal, to answer your proposal we have only added data related to the age of participants in each group.

2)     The present training period is 6 months. How long does it generally take before subject ongoing surgery? Would 6 months be representative? Would the effects also be present in much shorter periods?

Comment acknowledges. Preoperative period is different, not only among hospitals but also between patients. Patients who are waiting for bariatric surgery are asked for achieving a weight loss goal within a year generally, following protocolized psychological, nutritional and physical activity recommendations. If they do not reach this goal during this period, they are cleared off the bariatric surgery list.

We considered including a 6-months PAP added to the current protocol so as to analyse if it could help to achieve the weight loss goal during the 1-year period.

Regarding the effects presented in much shorter periods, that question is being addressed at the moment.

3)     The authors rightfully describe that caloric restriction may be useful to lose weight. Is there any nutritional data available? And have the authors considered implementing a nutritional intervention for this group?

Comment acknowledges. Although we don’t have the nutritional data, all the participants who collaborated in our research followed the same dietary recommendations given by their clinical teams. These recommendations were included in the clinical protocols for patients awaiting bariatric surgery and were the same for the EG and CG.

Regarding your last question, the aim of this study was to analyse how a PAP added to the current clinical protocols could be useful to help patients to lose body weight.

4)     The introduction is very long, I would advise to shorten the introduction to improve readability.

The authors appreciate the valuable comments provided by the reviewer, but at the same time, they think that is very complicated to contextualize the problem by reducing the introduction.

5)     There is an over presentation of the data. For example, lean body results are presented 3 times. In the table, individuals participant in a figure and in figure 3. The latter figure 3 I really like.

Comment acknowledges. We consider that each figure and table show different information.

Because of our small sample size, we think that Figure 2 is relevant to take into consideration how the same physical exercise intervention affects differently each participant at the individual level, since changes that we can see throughout the mean value (Table 3) are the result of the effects of the programme in every participant in EG, not observed in the CG.

We have modified tables 3 and 4 to improve their understanding, and part of the data have been added to an extra table sent as a supplementary material. The new tables 3 and 4 show within-group mean values and ES of anthropometry and physical condition variables separately, with the aim of demonstrating the absence of effects associated with the common clinical protocol in the CG, and the effects of our intervention in the EG.

At Figure 3, we show intergroup comparison.

6)     Is there any habitual physical activity data available. The latter is important to describe the energy expenditure which may explain the weight loss. What is the energy deficit that allows the ~9kg difference?

Comment acknowledges. Unfortunately, we had no accelerometers to register daily physical activity during the intervention. We’re aware of its relevance, since all of the components of energy balance interact with each other and affect total energy expenditure.

Regarding your question, we’re uncapable of answering completely. Weight loss could be considerably different between participants, which could be shown in Figure 2 through BMI values. The weight loss registered in Table 3 (~9 kg) is the mean value, but energy expenditure could be influenced by several variables such as energy intake, daily physical activity and basal metabolic rate. These limitations make us difficult to stablish a precise energy deficit as a cause of our differences in body weight. However, by the end of the programme, the participants accumulated 140 min/weekly of moderate-to-high aerobic exercise plus 2 sessions/weekly of resistance training, closely related to ACSM physical activity recommendations.

Round 2

Reviewer 1 Report

The authors have adequately addressed my comments. Differences are described as "among group" should be changed to "within group" as per scientific style.

Author Response

The authors are grateful for the contributions of the reviewer that have helped to improve the manuscript quality.

Also, English language and style were revised by a native specialist the first and second time we sent the manuscrit (we attach the proof-reading certificate). For that reason, we would be very grateful if you could tell us specifically which errors were detected.

Best regards

Reviewer 2 Report

The authors revised the manuscript as suggested by reviwers. 

Author Response

The authors are grateful for the contributions of the reviewer that have helped to improve the manuscript quality.

Best regards

Reviewer 3 Report

Fair answers and improved paper.

Author Response

The authors are grateful for the contributions of the reviewer that have helped to improve the manuscript quality.

The other two reviewers of the process have positively assessed the changes made during the review process in all the sections. The authors would appreciate that the reviewer could tell us what aspects of the introduction, research design and, methods can be improved.

Also, English language and style were revised by a native specialist the first and second time we sent the manuscrit (we attach the proof-reading certificate). For that reason, we would be very grateful if you could tell us specifically which errors were detected.

Best regards
